# Semi-Supervised Factored Logistic Regression for High-Dimensional Neuroimaging Data

**Danilo Bzdok, Michael Eickenberg, Olivier Grisel, Bertrand Thirion, Gaël Varoquaux**
INRIA, Parietal team, Saclay, France
CEA, Neurospin, Gif-sur-Yvette, France
firstname.lastname@inria.fr

## Abstract

Imaging neuroscience links human behavior to aspects of brain biology in ever-increasing datasets. Existing neuroimaging methods typically perform either discovery of unknown neural structure or testing of neural structure associated with mental tasks. However, testing hypotheses on the neural correlates underlying larger sets of mental tasks necessitates adequate representations for the observations. We therefore propose to blend representation modelling and task classification into a unified statistical learning problem. A multinomial logistic regression is introduced that is constrained by factored coefficients and coupled with an autoencoder. We show that this approach yields more accurate and interpretable neural models of psychological tasks in a reference dataset, as well as better generalization to other datasets.

**keywords**: Brain Imaging, Cognitive Science, Semi-Supervised Learning, Systems Biology

## 1 Introduction

Methods for neuroimaging research can be grouped by discovering neurobiological structure or assessing the neural correlates associated with mental tasks. To *discover*, on the one hand, spatial distributions of neural activity structure across time, independent component analysis (ICA) is often used [6]. It decomposes the BOLD (blood-oxygen level-dependent) signals into the primary modes of variation. The ensuing spatial activity patterns are believed to represent brain networks of functionally interacting regions [26]. Similarly, sparse principal component analysis (SPCA) has been used to separate BOLD signals into parsimonious network components [28]. The extracted brain networks are probably manifestations of electrophysiological oscillation frequencies [17]. Their fundamental organizational role is further attested by continued covariation during sleep and anesthesia [10]. Network discovery by applying ICA or SPCA is typically performed on task-unrelated (i.e., *unlabeled*) "resting-state" data. These capture brain dynamics during ongoing random thought without controlled environmental stimulation. In fact, a large portion of the BOLD signal variation is known not to correlate with a particular behavior, stimulus, or experimental task [10].

To *test*, on the other hand, the neural correlates underlying mental tasks, the general linear model (GLM) is the dominant approach [13]. The contribution of individual brain voxels is estimated according to a design matrix of experimental tasks. Alternatively, psychophysiological interactions (PPI) elucidate the influence of one brain region on another conditioned by experimental tasks [12]. As a last example, an increasing number of neuroimaging studies model experimental tasks by training classification algorithms on brain signals [23]. All these methods are applied to task-associated (i.e., *labeled*) data that capture brain dynamics during stimulus-guided behavior. Two important conclusions can be drawn. First, the mentioned supervised neuroimaging analyses typically yield results in a voxel space. This ignores the fact that the BOLD signal exhibits spatially distributed

patterns of coherent neural activity. Second, existing supervised neuroimaging analyses cannot exploit the abundance of easily acquired resting-state data [8]. These may allow better discovery of the manifold of brain states due to the high task-rest similarities of neural activity patterns, as observed using ICA [26] and linear correlation [9].

Both these neurobiological properties can be conjointly exploited in an approach that is *mixed* (i.e., using rest and task data), *factored* (i.e., performing network decomposition), and *multi-task* (i.e., capitalize on neural representations shared across mental operations). The integration of brain-network discovery into supervised classification can yield a semi-supervised learning framework. The most relevant neurobiological structure should hence be identified for the prediction problem at hand. Autoencoders suggest themselves because they can emulate variants of most unsupervised learning algorithms, including PCA, SPCA, and ICA [15, 16].

Autoencoders (AE) are layered learning models that condense the input data to local and global representations via reconstruction under compression prior. They behave like a (truncated) PCA in case of one linear hidden layer and a squared error loss [3]. Autoencoders behave like a SPCA if shrinkage terms are added to the model weights in the optimization objective. Moreover, they have the characteristics of an ICA in case of tied weights and adding a nonlinear convex function at the first layer [18]. These authors further demonstrated that ICA, sparse autoencoders, and sparse coding are mathematically equivalent under mild conditions. Thus, autoencoders may flexibly project the neuroimaging data onto the main directions of variation.

In the present investigation, a linear autoencoder will be fit to (unlabeled) rest data and integrated as a rank-reducing bottleneck into a multinomial logistic regression fit to (labeled) task data. We can then solve the compound statistical problem of unsupervised data representation and supervised classification, previously studied in isolation. From the perspective of dictionary learning, the first layer represents projectors to the discovered set of basis functions which are linearly combined by the second layer to perform predictions [20]. Neurobiologically, this allows delineating a low-dimensional manifold of brain network patterns and then distinguishing mental tasks by their most discriminative linear combinations. Theoretically, a reduction in model variance should be achieved by resting-state autoencoders that privilege the most neurobiologically valid models in the hypothesis set. Practically, neuroimaging research frequently suffers from data scarcity. This limits the set of representations that can be

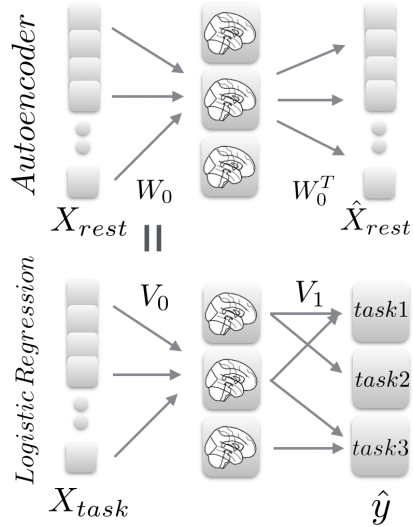

Figure 1: **Model architecture** Linear autoencoders find an optimized compression of 79,941 brain voxels into $n$ unknown activity patterns by improving reconstruction from them. The decomposition matrix equates with the bottleneck of a factored logistic regression. Supervised multi-class learning on task data ($X_{task}$) can thus be guided by unsupervised decomposition of rest data ($X_{rest}$).

extracted from GLM analyses based on few participants. We therefore contribute a computational framework that 1) analyzes many problems simultaneously (thus finds shared representations by "multi-task learning") and 2) exploits unlabeled data (since they span a space of meaningful configurations).

## 2 Methods

**Data.** As the currently biggest openly-accessible reference dataset, we chose resources from the Human Connectome Project (HCP) [4]. Neuroimaging task data with labels of ongoing cognitive processes were drawn from 500 healthy HCP participants (cf. Appendix for details on datasets). 18 HCP tasks were selected that are known to elicit reliable neural activity across participants (Table 1). In sum, the HCP task data incorporated 8650 first-level activity maps from 18 diverse paradigms administered to 498 participants (2 removed due to incomplete data). All maps were resampled to a common $60 \times 72 \times 60$ space of 3mm isotropic voxels and gray-matter masked (at least 10% tissue

probability). The supervised analyses were thus based on labeled HCP task maps with 79,941 voxels of interest representing z-values in gray matter.

| Cognitive Task | Stimuli | Instruction for participants |
|---|---|---|
| 1 Reward | Card game | Guess the number of a mystery card for gain/loss of money |
| 2 Punish | | |
| 3 Shapes | Shape pictures | Decide which of two shapes matches another shape geometrically |
| 4 Faces | Face pictures | Decide which of two faces matches another face emotionally |
| 5 Random | Videos with objects | Decide whether the objects act randomly or intentionally |
| 6 Theory of mind | | |
| 7 Mathematics | Spoken numbers | Complete addition and subtraction problems |
| 8 Language | Auditory stories | Choose answer about the topic of the story |
| 9 Tongue movement | Visual cues | Move tongue |
| 10 Food movement | | Squeezing of the left or right toe |
| 11 Hand movement | | Tapping of the left or right finger |
| 12 Matching | Shapes with textures | Decide whether two objects match in shape or texture |
| 13 Relations | | Decide whether object pairs differ both along either shape or texture |
| 14 View Bodies | Pictures | Passive watching |
| 15 View Faces | Pictures | Passive watching |
| 16 View Places | Pictures | Passive watching |
| 17 View Tools | Pictures | Passive watching |
| 18 Two-Back | Various pictures | Indicate whether current stimulus is the same as two items earlier |

Table 1: **Description of psychological tasks to predict.**

These labeled data were complemented by unlabeled activity maps from HCP acquisitions of unconstrained resting-state activity [25]. These reflect brain activity in the absence of controlled thought. In sum, the HCP rest data concatenated 8000 unlabeled, noise-cleaned rest maps with 40 brain maps from each of 200 randomly selected participants.

We were further interested in the utility of the optimized low-rank projection in one task dataset for dimensionality reduction in another task dataset. To this end, the HCP-derived network decompositions were used as preliminary step in the classification problem of another large sample. The ARCHI dataset [21] provides activity maps from diverse experimental tasks, including auditory and visual perception, motor action, reading, language comprehension and mental calculation. Analogous to HCP data, the second task dataset thus incorporated 1404 labeled, grey-matter masked, and z-scored activity maps from 18 diverse tasks acquired in 78 participants.

**Linear autoencoder.** The labeled and unlabeled data were fed into a linear statistical model composed of an autoencoder and dimensionality-reducing logistic regression. The affine autoencoder takes the input $\mathbf{x}$, projects it into a coordinate system of latent representations $\mathbf{z}$ and reconstructs it back to $\mathbf{x}'$ by

$$\mathbf{z} = \mathbf{W_0}\mathbf{x} + \mathbf{b_0} \qquad \mathbf{x}' = \mathbf{W_1}\mathbf{z} + \mathbf{b_1}, \qquad (1)$$

where $\mathbf{x} \in \mathbb{R}^{\mathbf{d}}$ denotes the vector of $d = 79,941$ voxel values from each rest map, $\mathbf{z} \in \mathbb{R}^{\mathbf{n}}$ is the $n$-dimensional hidden state (i.e., distributed neural activity patterns), and $\mathbf{x}' \in \mathbb{R}^{\mathbf{d}}$ is the reconstruction vector of the original activity map from the hidden variables. Further, $\mathbf{W_0}$ denotes the weight matrix that transforms from input space into the hidden space (encoder), $\mathbf{W_1}$ is the weight matrix for back-projection from the hidden variables to the output space (decoder). $\mathbf{b_0}$ and $\mathbf{b_1}$ are corresponding bias vectors. The model parameters $\mathbf{W_0}, \mathbf{b_0}, \mathbf{b_1}$ are found by minimizing the expected squared reconstruction error

$$\mathbb{E}\left[\mathcal{L}_{\mathcal{AE}}(\mathbf{x})\right] = \mathbb{E}\left[\|\mathbf{x} - (\mathbf{W_1}(\mathbf{W_0}\mathbf{x} + \mathbf{b_0}) + \mathbf{b_1})\|^2\right]. \qquad (2)$$

Here we choose $\mathbf{W_0}$ and $\mathbf{W_1}$ to be tied, i.e. $\mathbf{W_0} = \mathbf{W_1^T}$. Consequently, the learned weights are forced to take a two-fold function: That of signal *analysis* and that of signal *synthesis*. The first layer *analyzes* the data to obtain the cleanest latent representation, while the second layer represents building blocks from which to *synthesize* the data using the latent activations. Tying these processes together makes the analysis layer interpretable and pulls all non-zero singular values towards 1. Nonlinearities were not applied to the activations in the first layer.

**Factored logistic regression.** Our factored logistic regression model is best described as a variant of a multinomial logistic regression. Specifically, the weight matrix is replaced by the product

of two weight matrices with a common latent dimension. The later is typically much lower than the dimension of the data. Alternatively, this model can be viewed as a single-hidden-layer feed-forward neural network with a linear activation function for the hidden layer and a softmax function on the output layer. As the dimension of the hidden layer is much lower than the input layer, this architecture is sometimes referred to as a "linear bottleneck" in the literature. The probability of an input $\mathbf{x}$ to belong to a class $i \in \{1, \ldots, l\}$ is given by

$$P(Y = i|\mathbf{x}; \mathbf{V_0}, \mathbf{V_1}, \mathbf{c_0}, \mathbf{c_1}) = \mathrm{softmax}_i(f_{\mathcal{LR}}(\mathbf{x})), \tag{3}$$

where $f_{\mathcal{LR}}(\mathbf{x}) = \mathbf{V_1}(\mathbf{V_0}\mathbf{x} + \mathbf{c_0}) + \mathbf{c_1}$ computes multinomial logits and $\mathrm{softmax}_i(x) = \exp(x_i)/\sum_j \exp(x_j)$. The matrix $\mathbf{V_0} \in \mathbb{R}^{\mathbf{d \times n}}$ transforms the input $\mathbf{x} \in \mathbb{R}^{\mathbf{d}}$ into $n$ latent components and the matrix $\mathbf{V_1} \in \mathbb{R}^{\mathbf{n \times l}}$ projects the latent components onto hyperplanes that reflect $l$ label probabilities. $\mathbf{c_0}$ and $\mathbf{c_1}$ are bias vectors. The loss function is given by

$$\mathbb{E}\left[\mathcal{L}_{\mathcal{LR}}(\mathbf{x}, \mathbf{y})\right] \approx \frac{1}{N_{X_{task}}} \sum_{k=0}^{N_{X_{task}}} \log(P(Y = y^{(k)}|\mathbf{x^{(k)}}; \mathbf{V_0}, \mathbf{V_1}, \mathbf{c_0}, \mathbf{c_1})). \tag{4}$$

**Layer combination.** The optimization problem of the linear autoencoder and the factored logistic regression are linked in two ways. First, their transformation matrices mapping from input to the latent space are tied

$$\mathbf{V_0} = \mathbf{W_0}. \tag{5}$$

We hence search for a compression of the 79,941 voxel values into $n$ unknown components that represent a latent code optimized for both rest and task activity data. Second, the objectives of the autoencoder and the factored logistic regression are interpolated in the common loss function

$$\mathcal{L}(\theta, \lambda) = \lambda \mathcal{L}_{\mathcal{LR}} + (1 - \lambda) \frac{1}{N_{X_{rest}}} \mathcal{L}_{\mathcal{AE}} + \Omega. \tag{6}$$

In so doing, we search for the combined model parameters $\theta = \{\mathbf{V_0}, \mathbf{V_1}, \mathbf{c_0}, \mathbf{c_1}, \mathbf{b_0}, \mathbf{b_1}\}$ with respect to the (unsupervised) reconstruction error and the (supervised) task detection. $\mathcal{L}_{\mathcal{AE}}$ is devided by $N_{X_{rest}}$ to equilibrate both loss terms to the same order of magnitude. $\Omega$ represents an ElasticNet-type regularization that combines $\ell_1$ and $\ell_2$ penalty terms.

**Optimization.** The common objective was optimized by gradient descent in the SSFLogReg parameters. The required gradients were obtained by using the chain rule to backpropagate error derivatives. We chose the *rmsprop* solver [27], a refinement of stochastic gradient descent. *Rmsprop* dictates an adaptive learning rate for each model parameter by scaled gradients from a running average. The batch size was set to 100 (given much expected redundancy in $X_{rest}$ and $X_{task}$), matrix parameters were initalized by Gaussian random values multiplied by 0.004 (i.e., gain), and bias parameters were initalized to 0.

The normalization factor and the update rule for $\theta$ are given by

$$\begin{aligned}
\mathbf{v^{(t+1)}} &= \rho \mathbf{v^{(t)}} + (1 - \rho) \left(\nabla_\theta f(x^{(t)}, y^{(t)}, \theta^{(t)})\right)^2 \\
\theta^{(t+1)} &= \theta^{(t)} + \alpha \frac{\nabla_\theta f(x^{(t)}, y^{(t)}, \theta^{(t)})}{\sqrt{\mathbf{v^{(t+1)}} + \epsilon}},
\end{aligned} \tag{7}$$

where $f$ is the loss function computed on a minibatch sample at timestep $t$, $\alpha$ is the learning rate (0.00001), $\epsilon$ a global damping factor ($10^{-6}$), and $\rho$ the decay rate (0.9 to deemphasize the magnitude of the gradient). Note that we have also experimented with other solvers (stochastic gradient descent, adadelta, and adagrad) but found that *rmsprop* converged faster and with similar or higher generalization performance.

**Implementation.** The analyses were performed in Python. We used *nilearn* to handle the large quantities of neuroimaging data [1] and *Theano* for automatic, numerically stable differentiation of symbolic computation graphs [5, 7]. All Python scripts that generated the results are accessible online for reproducibility and reuse (`http://github.com/banilo/nips2015`).

# 3 Experimental Results

**Serial versus parallel structure discovery and classification.** We first tested whether there is a substantial advantage in combining unsupervised decomposition and supervised classification learning. We benchmarked our approach against performing data reduction on the (unlabeled) first half of the HCP task data by PCA, SPCA, ICA, and AE ($n = 5, 20, 50, 100$ components) and learning classification models in the (labeled) second half by ordinary logistic regression. PCA reduced the dimensionality of the task data by finding orthogonal network components (whitening of the data). SPCA separated the task-related BOLD signals into network components with few regions by a regression-type optimization problem constrained by $\ell_1$ penalty (no orthogonality assumptions, 1000 maximum iterations, per-iteration tolerance of $10^{-8}$, $\alpha = 1$). ICA performed iterative blind source separation by a parallel FASTICA implementation (200 maximum iterations, per-iteration tolerance of 0.0001, initialized by random mixing matrix, whitening of the data). AE found a code of latent representations by optimizing projection into a bottleneck (500 iterations, same implementation as below for rest data). The second half of the task data was projected onto the latent components discovered in its first half. Only the ensuing component loadings were submitted to ordinary logistic regression (no hidden layer, $\ell_1 = 0.1$, $\ell_2 = 0.1$, 500 iterations). These serial two-step approaches were compared against parallel decomposition and classification by SSFLogReg (one hidden layers, $\lambda = 1$, $\ell_1 = 0.1$, $\ell_2 = 0.1$, 500 iterations). Importantly, all trained classification models were tested on a large, unseen test set (20% of data) in the present analyses. Across choices for $n$, SSFLogReg achieved more than 95% out-of-sample accuracy, whereas supervised learning based on PCA, SPCA, ICA, and AE loadings ranged from 32% to 87% (Table 2). This experiment establishes the advantage of directly searching for classification-relevant structure in the fMRI data, rather than solving the supervised and unsupervised problems independently. This effect was particularly pronounced when assuming few hidden dimensions.

| $n$ | PCA + LogReg | SPCA + LogReg | ICA + LogReg | AE + LogReg | SSFLogReg |
|---|---|---|---|---|---|
| 5 | 45.1 % | 32.2 % | 37.5 % | 44.2 % | **95.7%** |
| 20 | 78.1 % | 78.2 % | 81.0 % | 63.2 % | **97.3%** |
| 50 | 81.7 % | 84.0 % | 84.2 % | 77.0 % | **97.6%** |
| 100 | 81.3 % | 82.2 % | 87.3 % | 76.6 % | **97.4%** |

Table 2: **Serial versus parallel dimensionality reduction and classification.** Chance is at 5,6%.

**Model performance.** SSFLogReg was subsequently trained (500 epochs) across parameter choices for the hidden components ($n = 5, 20, 100$) and the balance between autoencoder and logistic regression ($\lambda = 0, 0.25, 0.5, 0.75, 1$). Assuming 5 latent directions of variation should yield models with higher bias and smaller variance than SSFLogReg with 100 latent directions. Given the 18-class problem of HCP, setting $\lambda$ to 0 consistently yields generalization performance at chance-level (5,6%) because only the unsupervised layer of the estimator is optimized. At each epoch (i.e., iteration over the data), the out-of-sample performance of the trained classifier was assessed on 20% of unseen HCP data. Additionally, the "out-of-study" performance of the learned decomposition ($\mathbf{W_0}$) was assessed by using it as dimensionality reduction of an independent labeled dataset (i.e., ARCHI) and conducting ordinary logistic regression on the ensuing component loadings.

| | $n = 5$ | | | | | $n = 20$ | | | | | $n = 100$ | | | | |
|---|---|---|---|---|---|---|---|---|---|---|---|---|---|---|---|
| | $\lambda = 0$ | $\lambda = 0.25$ | $\lambda = 0.5$ | $\lambda = 0.75$ | $\lambda = 1$ | $\lambda = 0$ | $\lambda = 0.25$ | $\lambda = 0.5$ | $\lambda = 0.75$ | $\lambda = 1$ | $\lambda = 0$ | $\lambda = 0.25$ | $\lambda = 0.5$ | $\lambda = 0.75$ | $\lambda = 1$ |
| Out-of-sample accuracy | 6.0% | 88.9% | 95.1% | **96.5%** | 95.7% | 5.5% | 97.4% | **97.8%** | 97.3% | 97.3% | 6.1% | 97.2% | 97.0% | **97.8%** | 97.4% |
| Precision (mean) | 5.9% | 87.0% | 94.9% | **96.3%** | 95.4% | 5.1% | **97.4%** | 97.1% | 97.0% | 97.0% | 5.9% | 96.9% | 96.5% | **97.5%** | 96.9% |
| Recall (mean) | 5.6% | 88.3% | 95.2% | **96.6%** | 95.7% | 4.6% | **97.5%** | 97.5% | 97.4% | 97.4% | 7.2% | 97.2% | 97.2% | **97.9%** | 97.4% |
| F1 score (mean) | 4.1% | 86.6% | 94.9% | **96.4%** | 95.4% | 3.8% | **97.4%** | 97.2% | 97.1% | 97.1% | 5.3% | 97.0% | 96.7% | **97.7%** | 97.2% |
| Reconstr. error (norm.) | 0.76 | 0.85 | 0.87 | 1.01 | 1.79 | 0.64 | 0.67 | 0.69 | 0.77 | 1.22 | 0.60 | 0.65 | 0.68 | 0.73 | 1.08 |
| Out-of-study accuracy | 39.4% | 60.8% | 54.3% | 60.7% | **62.9%** | 77.0% | 79.7% | **81.9%** | 79.7% | 79.4% | 79.2% | **82.2%** | 81.7% | 81.3% | 75.8% |

Table 3: **Performance of SSFLogReg across model parameter choices.** Chance is at 5.6%.

We made three noteworthy observations (Table 3). First, the most supervised estimator ($\lambda = 1$) achieved in no instance the best accuracy, precision, recall, or f1 scores on HCP data. Classification by SSFLogReg is therefore facilitated by imposing structure from the unlabeled rest data. Confirmed by the normalized reconstruction error ($\mathbf{E} = \|x - \hat{x}\|/\|x\|$), little weight on the supervised term is sufficient for good model performance while keeping $\mathbf{E}$ low and task-map decomposition rest-like.

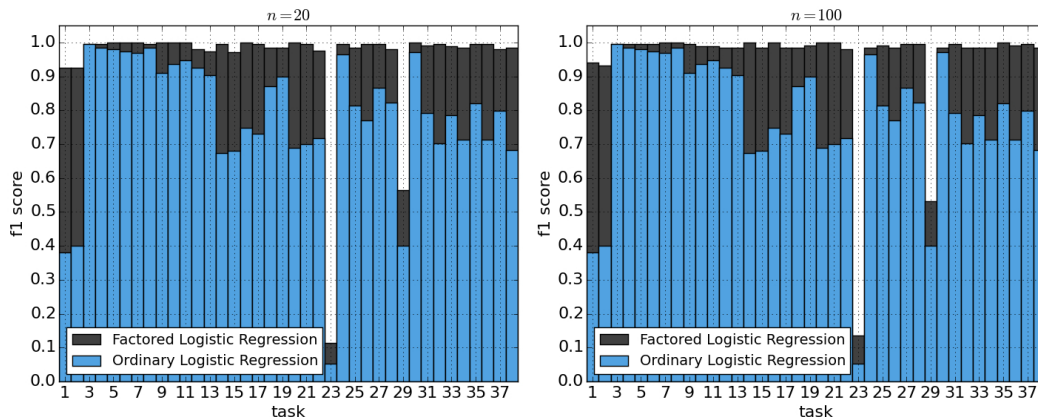

Figure 2: **Effect of bottleneck in a 38-task classificaton problem** Depicts the f1 prediction scores for each of 38 psychological tasks. Multinomial logistic regression operating in voxel space (*blue bars*) was compared to SSFLogReg operating in 20 (*left plot*) and 100 (*right plot*) latent modes (*grey bars*). Autoencoder or rest data were not used for these analyses ($\lambda = 1$). Ordinary logistic regression yielded 77.7% accuracy out of sample, while SSFLogReg scored at 94.4% ($n = 20$) and 94.2% ($n = 100$). Hence, compressing the voxel data into a component space for classification achieves higher task separability. Chance is at $2,6\%$.

Second, the higher the number of latent components $n$, the higher the out-of-study performance with small values of $\lambda$. This suggests that the presence of more rest-data-inspired hidden components results in more effective feature representations in unrelated task data. Third, for $n = 20$ and $100$ (but not 5) the purely rest-data-trained decomposition matrix ($\lambda = 0$) resulted in noninferior out-of-study performance of 77.0% and 79.2%, respectively (Table 3). This confirms that guiding model learning by task-unrelated structure extracts features of general relevance beyond the supervised problem at hand.

**Individual effects of dimensionality reduction and rest data.** We first quantified the impact of introducing a bottleneck layer disregarding the autoencoder. To this end, ordinary logistic regression was juxtaposed with SSFLogReg at $\lambda = 1$. For this experiment, we increased the difficulty of the classification problem by including data from all 38 HCP tasks. Indeed, increased class separability in component space, as compared to voxel space, entails differences in generalization performance of $\approx 17\%$ (Figure 2). Notably, the cognitive tasks on reward and punishment processing are among the least predicted with ordinary but well predicted with low-rank logistic regression (tasks 1 and 2 in Figure 2). These experimental conditions have been reported to exhibit highly similar neural activity patterns in GLM analyses of that dataset [4]. Consequently, also local activity differences (in the striatum and visual cortex in this case) can be successfully captured by brain-network modelling.

We then contemplated the impact of rest structure (Figure 3) by modulating its influence ($\lambda = 0.25, 0.5, 0.75$) in data-scarce and data-rich settings ($n = 20, \ell_1 = 0.1, \ell_2 = 0.1$). At the beginning of every epoch, 2000 task and 2000 rest maps were drawn with replacement from same amounts of task and rest maps. In data-scarce scenarios (frequently encountered by neuroimaging practitioners), the out-of-sample scores improve as we depart from the most supervised model ($\lambda = 1$). In data-rich scenarios, we observed the same trend to be apparent.

**Feature identification.** We finally examined whether the models were fit for purpose (Figure 4). To this end, we computed Pearson's correlation between the classifier weights and the averaged neural activity map for each of the 18 tasks. Ordinary logistic regression thus yielded a mean correlation of $\rho = 0.28$ across tasks. For SSFLogReg ($\lambda = 0.25, 0.5, 0.75, 1$), a per-class-weight map was computed by matrix multiplication of the two inner layers. Feature identification performance thus ranged between $\rho = 0.35$ and $\rho = 0.55$ for $n = 5$, between $\rho = 0.59$ and $\rho = 0.69$ for $n = 20$, and between $\rho = 0.58$ and $\rho = 0.69$ for $n = 100$. Consequently, SSFLogReg puts higher absolute weights on relevant structure. This reflects an increased signal-to-noise ratio, in part explained by

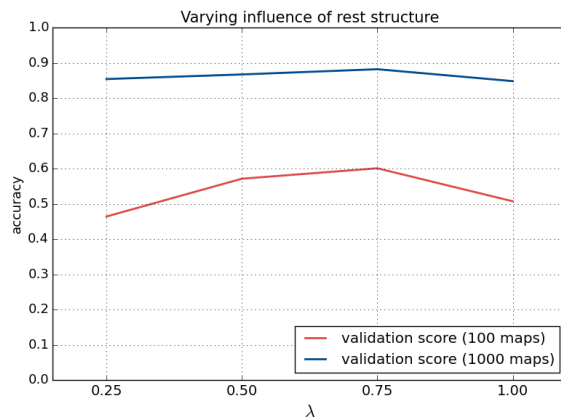

Figure 3: **Effect of rest structure** Model performance of SSFLogReg ($n = 20$, $\ell_1 = 0.1$, $\ell_2 = 0.1$) for different choices of $\lambda$ in data-scarce (100 task and 100 rest maps, *hot color*) and data-rich (1000 task and 1000 rest maps, *cold color*) scenarios. Gradient descent was performed on 2000 task and 2000 rest maps. At the begining of each epoch, these were drawn with replacement from a pool of 100 or 1000 different task and rest maps, respectively. Chance is at 5.6%.

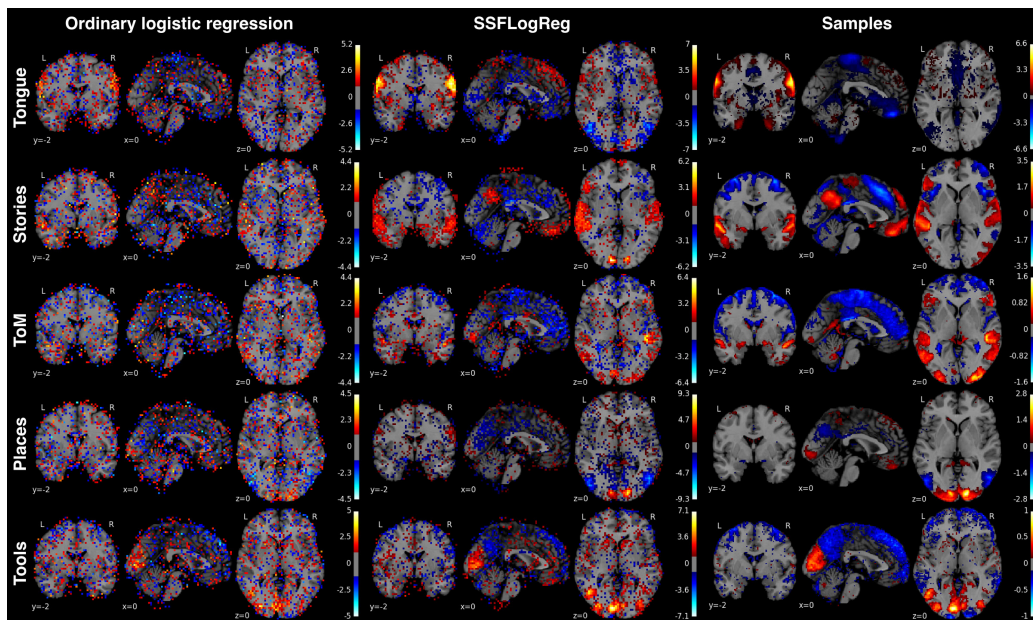

Figure 4: **Classification weight maps** The voxel predictors corresponding to 5 exemplary (of 18 total) psychological tasks (*rows*) from the HCP dataset [4]. *Left column:* multinomial logistic regression (same implementation but without bottleneck or autoencoder), *middle column:* SSFLogReg ($n = 20$ latent components, $\lambda = 0.5$, $\ell_1 = 0.1$, $\ell_2 = 0.1$), *right column:* voxel-wise average across all samples of whole-brain activity maps from each task. SSFLogReg a) puts higher absolute weights on relevant structure, b) lower ones on irrelevant structure, and c) yields BOLD-typical local contiguity (without enforcing an explicit spatial prior). All values are z-scored and thresholded at the $75^{th}$ percentile.

the more BOLD-typical local contiguity. Conversely, SSFLogReg puts lower probability mass on irrelevant structure. Despite lower interpretability of the results from ordinary logistic regression, the salt-and-pepper-like weight maps were sufficient for good classification performance. Hence, SSFLogReg yielded class weights that were much more similar to features of the respective training samples for all choices of $n$ and $\lambda$. SSFLogReg therefore captures genuine properties of task activity patterns, rather than participant- or study-specific artefacts.

**Miscellaneous observations.** For the sake of completeness, we informally report modifications of the statistical model that did not improve generalization performance. $a$) Introducing stochasticity into model learning by input corruption of $\mathbf{X_{task}}$ deteriorated model performance in all scenarios. Adding $b$) rectified linear units (ReLU) to $\mathbf{W_0}$ or other commonly used nonlinearities ($c$) sigmoid, $d$) softplus, $e$) hyperbolic tangent) all led to decreased classification accuracies, probably due to sample size limits. Further, $f$) "pretraining" of the bottleneck $\mathbf{W_0}$ (i.e., non-random initialization) by either corresponding PCA, SPCA or ICA loadings did not exhibit improved accuracies, neither did $g$) autoencoder pretraining. Moreover, introducing an additional $h$) overcomplete layer (100 units) after the bottleneck was not advantageous. Finally, imposing either $i$) only $\ell_1$ or $j$) only $\ell_2$ penalty terms was disadvantageous in all tested cases. This favored ElasticNet regularization chosen in the above analyses.

## 4    Discussion and Conclusion

Using the flexibility of factored models, we learn the low-dimensional representation from high-dimensional voxel brain space that is most important for prediction of cognitive task sets. From a machine-learning perspective, factorization of the logistic regression weights can be viewed as transforming a "multi-class classification problem" into a "multi-task learning problem". The higher generalization accuracy and support recovery, comparing to ordinary logistic regression, hold potential for adoption in various neuroimaging analyses. Besides increased performance, these models are more interpretable by automatically learning a mapping to and from a brain-network space. This domain-specific learning algorithm encourages departure from the artificial and statistically less attractive voxel space. Neurobiologically, brain activity underlying defined mental operations can be explained by linear combinations of the main activity patterns. That is, fMRI data probably concentrate near a low-dimensional manifold of characteristic brain network combinations. Extracting fundamental building blocks of brain organization might facilitate the quest for the cognitive primitives of human thought. We hope that these first steps stimulate development towards powerful semi-supervised representation extraction in systems neuroscience.

In the future, automatic reduction of brain maps to their neurobiological essence may leverage data-intense neuroimaging investigations. Initiatives for data collection are rapidly increasing in neuroscience [22]. These promise structured integration of neuroscientific knowledge accumulating in databases. Tractability by condensed feature representations can avoid the ill-posed problem of learning the full distribution of activity patterns. This is not only relevant to the multi-class challenges spanning the human cognitive space [24] but also the multi-modal combination with high-resolution 3D models of brain anatomy [2] and high-throughput genomics [19]. The biggest socioeconomic potential may lie in across-hospital clinical studies that predict disease trajectories and drug responses in psychiatric and neurological populations [11].

**Acknowledgment**    The research leading to these results has received funding from the European Union Seventh Framework Programme (FP7/2007-2013) under grant agreement no. 604102 (Human Brain Project). Data were provided by the Human Connectome Project. Further support was received from the German National Academic Foundation (D.B.) and the MetaMRI associated team (B.T., G.V.).

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
