[Supplementary Material]

# 5 Appendix

**Data.**    As the currently biggest openly-accessible reference dataset, we chose the Human Connectome Project (HCP) resources [4]. Neuroimaging task data with labels of ongoing cognitive processes were drawn from 500 healthy HCP participants. 18 HCP tasks were selected that are known to elicit reliable neural activity across participants. The task paradigms include 1) working memory/cognitive control processing, 2) incentive processing, 3) visual and somatosensory-motor processing, 4) language processing (semantic and phonological processing), 5) social cognition, 6) relational processing, and 7) emotional processing. All data were acquired on the same Siemens Skyra 3T scanner. Whole-brain EPI acquisitions were acquired with a 32 channel head coil (TR=720ms, TE=33.1ms, flip angle=52, BW=2290Hz/Px, in-plane FOV=280mm $\times$ 180mm, 72 slices, 2.0mm isotropic voxels). The "minimally preprocessed" pipeline includes gradient unwarping, motion correction, fieldmap-based EPI distortion correction, brain-boundary-based registration of EPI to structural T1-weighted scans, nonlinear (FNIRT) registration into MNI space, and grand-mean intensity normalization. Activity maps were spatially smoothed with a Gaussian kernel of 4mm (FWHM). A GLM was implemented by FILM from the FSL suite with model regressors from convolution with a canonical hemodynamic response function and from temporal derivatives. HCP tasks were conceived to modulate activity in a maximum of different brain regions and neural systems. Indeed, at least 70% of the participants showed consistent brain activity in contrasts from the task battery, which certifies excellent activity patterns covering extended parts of the brain [4]. In sum, the HCP task data incorporated 8650 first-level activity maps from 18 diverse paradigms administered to 498 participants (2 removed due to incomplete data). All maps were resampled to a common 60x72x60 space of 3mm isotropic voxels and gray-matter masked (at least 10% tissue probability). The supservised analyses were based on labeled HCP task maps with 79,941 voxels of interest representing z-values in gray matter.

These labeled data were complemented by unlabeled activity maps from HCP acquisitions of unconstrained resting-state activity [25]. These reflect brain activity in the absence of controlled thought. In line with the goal of the present study, acquisition of these data was specifically aimed at the study of task-rest correspondence. From each participant, we included two time-series for left and right phase encoding with 1,200 maps of multiband, gradient-echo planar imaging acquired during a period of 15min (TR=720 ms, TE=33.1 ms, flip angle=52, FOV=280mm $\times$ 180mm, and 2.0mm isotropic voxels). Besides run duration, the task acquisitions were identical to the resting-state fMRI acquisitions for maximal compatibility between task and rest data. We here drew on "minimally preprocessed" rest data from 200 randomly selected healthy participants. PCA was applied to each set of 1,200 rest maps for denoising by keeping only the 20 main modes of variation. In sum, the HCP rest data concatenated 8000 unlabeled, noise-cleaned rest maps with 40 brain maps from each of 200 randomly selected participants.

We further evaluated whether the low-dimensional space learned in HCP task/rest data can be re-used as a feature extraction step for learning classification models in an independent task dataset. These experiments therefore probe the generality of the learned representation by assessing transfer learning effects. To this end, the HCP-derived network decompositions were used as preliminary step in the classification problem of another large sample. The ARCHI dataset [21] provides activity maps from diverse experimental tasks, including auditory and visual perception, motor action, reading, language comprehension and mental calculation. 81 right-handed healthy participants (3 not included in present analyses due to incomplete data) without psychiatric or neurological history participated in four fMRI sessions acquired under different experimental paradigms. The functional maps were warped into the MNI space and resampled to isotropic 3mm resolution. Whole-brain EPI data were acquired with the same Siemens Trio with a 32 channel head coil (TR=2400ms, TE=30ms, flip angle=60, in-plane FOV=192mm $\times$ 192mm, 40 slices, 3.0mm isotropic voxels). Standard preprocessing was performed with Nipype [14], including slice timing, motion correction, alignment, and spatial normalization. Activity maps were spatially smoothed by a Gaussian kernel of 5mm (FWHM). Analogous to HCP data, the second task dataset incorporated 1404 labeled, grey-matter masked, and z-scored activity maps from 18 diverse tasks acquired in 78 participants.

Figure 5: **Weight maps of a same hidden factor ranging from unsupervised to supervised regime** One of the $n$ factors from the hidden layer ($\mathbf{W_0}$) was plotted for the same data (full HCP dataset) and the same model choices ($n = 20, \ell_1 = 0.1, \ell_2 = 0.1$) along a $\lambda$-grid between purely unsupervised ($\lambda = 0.0$, *top row*) and purely supervised ($\lambda = 1.0$, *bottom row*) settings. As qualitative evidence, a slow transition from rest- to task-typical brain networks was observed in brain space. Although difficult to quantify, rest network elements appear to get 'reassembled' to latent factors of the LR. This increased confidence that the improved model performance of rest-informed fLR is not only an arbitrary effect of spatially smooth noise. All values are z-scored.