[Reviews · NeurIPS 2015]

Submitted by Assigned_Reviewer_1

The paper combines unsupervised learning on rest-state fMRI data (using a linear autoencoder) with supervised learning on task-based fMRI data (factored linear regression on task-based fMRI data). The combination is done by sharing the first matrix and using an objective which is a linear combination between reconstruction error (autoencoder) and classification error (regression).

Comments:

The paper reports a surprising result: The combined method obtains an out-of-sample accuracy of greater or equal than 95% whereas the best chain of independent steps for dimensionality reduction and regression achieves 87% or less. This is potentially a very interesting finding but missing information in the manuscript make it hard to understand the reason for this result.

Specific comments:

1) Eq. 4 is missing a closing parenthesis and, more importantly, the rationale behind this objective for regression needs to be explained. A potential problem with this objective is that it diverges to minus infinity if one class membership probability is exactly zero.

2) It should be explained why \lambda = 1 is chosen for table 2 and in parts of table 3. For this setting the objective does not at all reflect reconstruction error - basically only regression performance is optimized.

3) It would be good to also report the reconstruction performances achieved by the different models.
Summary: The paper reports a surprising result: The combined method obtains an out-of-sample accuracy of greater or equal than 95% whereas the best chain of independent steps for dimensionality reduction and regression achieves 87% or less. This is potentially a very interesting finding but missing information in the manuscript makes it hard to understand the reason for this result.

Submitted by Assigned_Reviewer_2

This paper combines resting state data analysis by a linear auto-encoder and multitask learning in a multinomial logistic regression model with factored coefficients. It uses Human Connectome Project data, off-the-shelf tools (gradient descent optimization), 18 tasks for training.

Again, this sort of combination

of an auto-encoder and task-specific supervised learning goes back to at least Hertz, Krogh & Palmer (1991).

The fact that combining the two methods has an advantage over not doing it is well known, albeit it is nice that the authors demonstrated this once again with large-scale fMRI data.

What I am missing is a deeper analysis of why the compressing helps at all.

Is it just pure luck, or the resting state data indeed carries some functionally significant information beyond simply reducing the search space to continuous patches? There are a number of papers in the literature elaborating on the concept of how resting-state activity might be revealing about the functional organization of the cortex.

A link to this literature, a test or analysis would have been nice.

A minimal control experiment that I would have liked to see is adding a completely arbitrary but spatially compact regularizer to the auto-encoder and see how much improvement the system can achieve..
Summary: A potentially interesting idea implemented with standard tools, but also a missed opportunity for a deeper work

Author Feedback
Author rebuttal: To Reviewer 2/4/5/7:
The neuroimaging field is still limited to *either* using unsupervised methods to find "resting-state networks" (PCA,SparsePCA,ICA,etc.) *or* using supervised methods on experimental data to predict psychological processes (logistic regression,SVMs,etc.). Yet, recent neuroimaging studies evidence a close correspondence between resting-state correlations and task brain activity (Smith et al., 2009 PNAS; Cole et al., 2014 Neuron). The unsupervised discovery of special structure and supervised prediction of mental tasks are unfortunately seldom combined in brain imaging. Determining predictive brain regions of defined mental operations could be facilitated by learning in the modes of variation most relevant to the multi-task problem. We thus integrated flexible brain-network discovery by autoencoders (AEs) applied to rest data and mental task prediction by factored logistic regression (fLR) applied to task data.
First, the proposed AE/fLR-hybrid classified very well 18 psychological tasks from the HCP dataset (out-of-sample accuracies with n=5/20/50/100 components: 95,7-97,6% [range]); much better than the *serial* approach with initial PCA/SparsePCA/ICA/AE reduction of rest data and subsequent LR task classification on component loadings (out-of-sample accuracies: 32,2-87,3% [range]). Second, benefitting from rest structure improved the supervised map decomposition with out-of-dataset performances from 79,4% to 81,9% (n=20) and from 75,8% to 82,2% (n=100) in 18 *unseen* tasks (ARCHI). Third, the integrated AE/fLR considerably enhanced feature identification from r=0.28 (ordinary LR) up to r=0.55 (n=5), r=0.69 (n=20), and r=0.69 (n=100) [voxelwise Pearson correlation between model weights and class maps]. Exploiting rest and task data by the AE/fLR-model thus improved the predictive weight maps for interpretability by cognitive neuroscientists, besides objective model performance gains.
In sum, recent neuroscientific evidence for similar neural activity patterns during task and rest (Smith/2009/PNAS; Cole/2014/Neuron) were incorporated into a domain-specific learning model for joint dimensionality reduction and mental task classification. It enables mapping of traditional psychological concepts on combinations of brain networks shared across diverse tasks.

Reviewer 2/point 1:
Hertz et al., 1991 will be included in the revised introduction.

Reviewer 2/point 2:
Evidence for the usefulness of rest data is threefold:
1) The purely supervised fLR model (no AE/rest, lambda=1) achieved in no instance the best accuracy, precision, recall or f1 scores on HCP data across component choices (n=5/20/100).
2) The decomposition matrix from purely unsupervised AE learning on rest data (lambda=0) proved useful for data reduction in classifying 18 independent tasks (ARCHI) with 77% and 79,2% accuracy instead of max. 81,9% and 82,2% (n=20/100) with supervised decomposition matrices.
3) Rest-informed classification improved the out-of-sample performance by 10% in data-scarcity (100/100 task/rest maps) and 2% in data-richness (1000/1000).

Reviewer 2/point 3:
We will acknowledge specific previous neuroimaging studies on task-rest architecture by citing+explaining Smith et al., 2009 and Cole et al., 2014 (cf. above) in the introduction.

Reviewer 5/point 1:
The AE layer was examined with/without nonlinearities (ReLU, sigmoid, softmax, tanh), L1 and L2 penalties; but linearity with small L1+L2 terms works best.
Yes, linear+L1+L2 AE equates with a sparse PCA, yet its encoding matrix (W0) is shared with fLR (V0). This AE layer performs better than ICA and than PCA without L1 and/or L2.

Reviewer 7/point 1:
Brackets added.
Our LR objective was motivated by a widely used form: http://deeplearning.net/tutorial/logreg.html

Reviewer 7/point 2:
Table 2 used fLR without AE (lambda=1) to demonstrate the superiority of parallel over serial structure discovery+classification, intentionally disregarding rest data.
Table 3 reports AE/fLR model behavior along a coarse grid of n and lambda choices.

Reviewer 7/point 3:
As new result for Table 3, the reconstruction error was computed as E=||X-Xrec|| / ||X|| (0=best,>1=bad): E=0.76/0.85/0.87/1.01/1.79 for n=5 (lambda=0/0.25/0.5/0.75/1), E=0.64/0.67/0.69/0.77/1.22 for n=20 and E=0.60/0.65/0.68/0.73/1.08 for n=100. Thus, little weight on the LR term is sufficient for good model performance while keeping E low and task-map decomposition rest-like. Training these models, AE and LR losses decreased continuously over 500 minibatch iterations, except for lambda=1.0 (no AE).

As qualitative evidence, plotting the decomposition matrix V0/W0 from lambda=0 to 1 in brain space slowly transitioned from rest- to task-typical brain networks. Although difficult to quantify, rest network elements are "reassembled" to latent factors of the LR. This increased our confidence against an arbitrary effect of spatially smooth noise. We'll add examples to the appendix.